# Linear Proportional-Integral-Differential-Robustified Continuous-Time Optimal Predictive Control for a Class of Nonlinear Systems

**Guilin Wu** [1,*], **Haisong Ang** [1] and **Qingxi Li** [2]

[1]   College of Aerospace Engineering, Nanjing University of Aeronautics and Astronautics,
    Nanjing 211106, China; anghaisong@nuaa.edu.cn
[2]   Avic General Aircraft Research Institute, Zhuhai 519042, China; liqx023@avic.com
[*]   Correspondence: guilinwu@nuaa.edu.cn

**Abstract:** This paper presents a novel robust optimal predictive control approach for a class of nonlinear continuous-time systems perturbed by unknown disturbances. First, a new error state with a linear proportional-integral-differential (PID) structure considering current, accumulative, and derivative tracking errors is defined. Second, prediction of the error state within the predictive periods is expressed by the error state and its high-order derivatives according to the Taylor series expansion. Last, the proposed control law as well as the main result of this paper are derived by minimizing the prediction of the error state. Numerical validation for designing a missile autopilot shows that, due to minimizing the accumulative tracking error included in the PID-structuralized new error state, the proposed approach can generate smaller steady-state tracking errors than two commonly applied continuous-time optimal predictive control approaches whether the disturbances encountered by the missile are constant or time-varying.

**Keywords:** optimal predictive control; nonlinear continuous-time systems; PID; unknown disturbances; steady-state tracking error

## 1. Introduction

Control design for nonlinear systems has always been a challenging task. Generally, the control approaches can be classified into linear and nonlinear catalogs. In the linear approaches, by simplifying or even neglecting partial nonlinearities, the systems are linearized around equilibrium points such that the classical and modern linear control theories [1,2] are applicable. However, in order to give consideration of the nonlinearities as much as possible to meet the increasing demand of high-quality control performances, various nonlinear control approaches such as the feedback linearization [3], the sliding mode control [4], the adaptive methodology [5], the finite time control [6], the backstepping approach [7], and so forth have been proposed and applied.

As a promising and viable nonlinear control approach, the continuous-time optimal predictive control (CTOPC) methodology [8] extends a novel prospect for the control design of the nonlinear systems. In the prototype CTOPC (PCTOPC), state of the dynamical system or the prescribed reference is predicted by a linear combination of its high-order derivatives by means of the Taylor series expansion. Then, the optimal predictive control law is obtained by minimizing a quadratic receding horizon performance index (RHPI), which is the function of the predictive response errors. The control law has the merit of giving full consideration of all the system nonlinearities due to the complete inclusion of the system dynamics, which makes the PCTOPC more attractive and have many successful applications in various fields during the past decades such as the flight control [9,10], the permanent magnet synchronous motor control [11], angle tracking control of a steer-by-wire system [12], the DC-DC converter control [13–15], the photovoltaic inverter control [16,17],

the wind energy system design [18,19], the water tank system design [20,21], the active suspension system design [22], stabilization of an artificial gas-lift process [23], path following system design of autonomous mobile vehicles [24,25], and solar power plant control [26]. Though the PCTOPC has drawn much attention in the aforementioned areas, its inadequacy is also obvious. That is, the closed-loop system is susceptible to the unknown external uncertainties since the PCTOPC only takes into account of the current tracking errors between the prescribed reference and the system output, making poor robustness and large steady-state tracking errors easy to occur. Thus far, only few previous works have been dedicated to address such problems [9,19,21,23–26]. Therefore, targeting the inadequacy and extending an alternative methodology to enhance the robustness of the PCTOPC are meaningful and of engineering significance.

As one of the most widely used approaches around the world, the linear proportional-integral-differential (PID), which not only has the merits of simple controller structure and easy implementation procedures, but also has the strong capability on dealing with the unknown disturbances, has many successful application examples in various fields [27–29]. In the PID, considering the terms of accumulative tracking errors can help the closed-loop systems reduce stead-state errors. By assigning appropriate proportional, integral, and differential gain coefficients, satisfactory system response time and input performance can be acquired. Hence, the PID approach has been successfully applied to enhance the system robustness for the existing approaches [30,31]. Due to those advantages, this paper combines the PCTOPC with the linear PID approach for closed-loop system robustness enhancement. Next, the proposed approach is named as PID-CTOPC.

To further highlight superiorities of the proposed approach, comparison with another typical CTOPC approach [32,33], which considers both current and high-derivative tracking errors proposed by Professor Lu, is also conducted. The main contributions of this paper are twofold.

1.  A closed-loop steady-state tracking error reduction approach is proposed in the field of CTOPC. The accumulative tracking errors, which are integrated into a new error state with PID structure, are considered in the proposed PID-CTOPC approach such that tracking errors within the operation envelop of the closed-loop system can be minimized compared with the PCTOPC and the Lu's approach, which only consider the current tracking errors.
2.  Robustness of the closed-loop system is significantly enhanced. In the proposed PID-CTOPC approach, current, accumulative, and derivative tracking errors are minimized simultaneously such that reduction of input fluctuation, output overshoot, and steady-state tracking error can also be derived in contrast with the PCTOPC and the Lu's approach in the presence of unknown disturbances.

## 2. Preliminaries

In this section, some useful preliminaries such as symbols, lemmas, and existing optimal predictive approaches are introduced first.

Denote $\mathbb{Z}^+$ as the set of positive integers, $\mathbb{Z} = \{0, \mathbb{Z}^+\}$, $\mathbb{R}$ as the set of real numbers, $\mathbb{R}^{p \times q}$ as the set of $p \times q$ real matrices, $\mathbb{R}^s$ as the set of column matrices with dimension $s$, $O^{m \times n}$ as the set of $m \times n$ zero matrices, $s^{[j]}$ as the $j$th derivative of $s$ relative to time, $s!$ as the factorial of $s \in \mathbb{Z}$.

This paper considers the following single-input–single-output (SISO) nonlinear affine systems:

$$\begin{cases} \dot{x}_i(t) = x_{i+1}(t), \ i = 1, \dots, n-1 \\ \dot{x}_n(t) = f(x(t)) + g(x(t)) \cdot u(t) \\ y(t) = x_1(t) \end{cases} \tag{1}$$

where, $x(t) = [x_1(t), \dots, x_n(t)] \in \mathbb{R}^n$ represents the state vector, $n \geq 2$ represents the system dimension. $y(t) \in \mathbb{R}$ and $u(t) \in \mathbb{R}$ represent output and input, respectively. $f(x(t)) \in \mathbb{R}$ and $g(x(t)) \in \mathbb{R}$ are nonlinear mappings with $g(x) \neq 0$ for any $x(t)$. In next,

the variable $\vartheta(t)$ is denoted as $\vartheta$ while the prediction of $\vartheta$ within the predictive period $\tau > 0$ would be expressed by $\vartheta(t + \tau)$, $\vartheta$ is a scalar or vector. The following assumptions are imposed on system (1):

(A1) The zero dynamics are stable.
(A2) All the states $x_i$, $i = 1, \ldots, n$ are available.
(A3) The system output $y$ and the prescribed reference $y_d$ are sufficiently many times continuously differentiable relative to time $t$.

Secondly, the PCTOPC approach is reviewed.

Differentiating both sides of the last state differential equation in Formula (1) $\sigma$ times relative to $t$ yields:

$$
\begin{cases}
y^{[n+1]} = p_1\left(u^2, u; x\right) + g(x) \cdot \dot{u} \\
y^{[n+2]} = p_2\left(u^3, u^2, u, \dot{u}; x\right) + g(x) \cdot \ddot{u} \\
\quad \vdots \\
y^{[n+\sigma]} = p_\sigma\left(u^{\sigma+1}, \ldots, u, u^{[\sigma-1]}, \ldots, \dot{u}; x\right) + g(x) \cdot u^{[\sigma]}
\end{cases}
\tag{2}
$$

where, $\sigma \in \mathbb{Z}^+$ is called control order. Details of the nonlinear terms $p_i$, $i = 1, \ldots, \sigma$ can be seen in [8].

By using the Taylor series expansion, predictions of $y$ and $y_d$ within the predictive period $t_C$ can be expressed by:

$$
\begin{cases}
y(t + \tau) = \Gamma_c \cdot Y_c \\
y_d(t + \tau) = \Gamma_c \cdot Y_{cd}
\end{cases}
\tag{3}
$$

where, $0 < \tau \leq t_C$, $\Gamma_c = \left[1, \tau, \ldots, \frac{\tau^{n+\sigma}}{(r+\sigma)!}\right]$, $Y_c(t) = \left[y, \dot{y}, \ldots, y^{[n+\sigma]}\right]^T$, and $Y_{cd}(t) = \left[y_d, \dot{y}_d, \ldots, y_d^{[n+\sigma]}\right]^T$.

By employing following RHPI:

$$
J_C = \frac{1}{2}\int_0^{t_C}[y(t + \tau) - y_d(t + \tau)]^T[y(t + \tau) - y_d(t + \tau)]d\tau
\tag{4}
$$

the optimal predictive control law for the system (1) is summarized in following lemma.

**Lemma 1** [8]. *Consider the nonlinear affine system (1) satisfying (A1)~(A3). Then, for a given control order $\sigma \in \mathbb{Z}^+$, the system output $y$ and the prescribed reference $y_d$ are predicted by Formula (3). The optimal predictive control law minimizing the RHPI (4) is given by:*

$$
u(t) = [g(x)]^{-1}\left[y_d^{[n]} - f(x) - K_c M_r\right]
\tag{5}
$$

*where, $M_r$ is given by:*

$$
M_r = \left[x_1 - y_d, x_2 - \dot{y}_d, \ldots, x_n - y_d^{[n-1]}\right]^T
\tag{6}
$$

$K_c \in \mathbb{R}^{1 \times (\sigma+1)}$ *is the first row of matrix $A_\sigma^{-1} A_1$ . The matrices $A_1$ and $A_\sigma$ are given by:*

$$
\begin{cases}
A = \int_0^{t_1} \Gamma_c^T \Gamma_c d\tau = \begin{bmatrix} A_r & A_1^T \\ A_1 & A_\sigma \end{bmatrix} \in \mathbb{R}^{(n+\sigma+1) \times (n+\sigma+1)} \\
A_r \in \mathbb{R}^{n \times n}, \quad A_\sigma \in \mathbb{R}^{(\sigma+1) \times (\sigma+1)}
\end{cases}
\tag{7}
$$

### 3. The PID-CTOPC Approach

This section presents main results of this paper. Firstly, a new error state with PID structure is introduced, transforming the prescribed reference tracking problem into a stabilization one for the system (1). Secondly, the robustified optimal predictive controller is designed.

*3.1. Stabilization Model*

A new error state with PID structure is defined as follows:

$$s(t) = K_p e_1 + K_i \int_0^t e_1 dt_0 + K_d \dot{e}_1 \tag{8}$$

where, $e_1 = x_1 - y_d$. $K_p$, $K_i$, and $K_d$ are tuning parameters, representing proportional, integral, and differential coefficients, respectively. In the above, $\int_0^t e_1 dt_0$ represents the accumulative error and $\dot{e}_1$ represents the derivative error.

Differentiating $(n-1)$ times for the new state $s(t)$ relative to $t$ yields:

$$\begin{cases} \dot{s} = K_p \dot{e}_1 + K_i e_1 + K_d \ddot{e}_1 \\ \quad \vdots \\ s^{[n-2]} = K_p e_1^{[n-2]} + K_i e_1^{[n-3]} + K_d e_1^{[n-1]} \\ s^{[n-1]} = K_p e_1^{[n-1]} + K_i e_1^{[n-2]} + K_d e_1^{[n]} \end{cases} \tag{9}$$

Then, by taking into account of Formula (1), $s^{[n-1]}$ can be written into the following affine form:

$$\begin{aligned} s^{[n-1]} &= \underbrace{K_p e_1^{[n-1]} + K_i e_1^{[n-2]} + K_d \left[ f(x) - y_d^{[n]} \right]}_{\overline{f}(x)} + \underbrace{K_d \cdot g(x)}_{\overline{g}(x)} \cdot u \\ &= \overline{f}(x) + \overline{g}(x) \cdot u \end{aligned} \tag{10}$$

Since the objectives of controlling system (1) are $x_1 \to y_d$ and $x_i \to y_d^{[i-1]}$, $i = 2, \ldots, n$, hence, the new error state given by Formula (8) is expected to converge to zero, namely, $s \to 0$, which implies that the prescribed reference tracking problem of system (1) is transformed into a stabilization one.

*3.2. Controller Design*

In Formula (10), differentiating $\rho$ times for $s^{[n-1]}$ relative to $t$ yields:

$$\begin{cases} s^{[n]} = \phi_1 \left( u^2, u; x \right) + \overline{g}(x) \cdot \dot{u} \\ s^{[n+1]} = \phi_2 \left( u^3, u^2, u, \dot{u}; x \right) + \overline{g}(x) \cdot \ddot{u} \\ \quad \vdots \\ s^{[n-1+\rho]} = \phi_\rho \left( u^{\rho+1}, \ldots, u, u^{[\rho-1]}, \ldots, \dot{u}; x \right) + \overline{g}(x) \cdot u^{[\rho]} \end{cases} \tag{11}$$

where, $\phi_i \left( u^{i+1}, \ldots, u, u^{[i-1]}, \ldots, \dot{u}; x \right)$, $i = 1, \ldots, \rho$ are nonlinear functions. $\rho$ represents the control order. Let:

$$\begin{cases} \overline{S} = \left[ H_0^T, H^T \right]^T \\ H_0 = \left[ s, \dot{s}, \ldots, s^{[n-2]} \right]^T \\ H = \left[ s^{[n-1]}, s^{[n]}, \ldots, s^{[n-1+\rho]} \right]^T \end{cases} \tag{12}$$

Then, prediction of $s(t)$ within the predictive period $t_M$ can be written as:

$$s(t + \tau) = \Gamma_M \overline{S} \tag{13}$$

where, $0 < \tau \leq t_M$ and $\Gamma_M = \left[1, \tau, \ldots, \frac{\tau^{n+\rho-1}}{(n+\sigma-1)!}\right] \in \mathbb{R}^{1 \times (n+\rho)}$.

The RHPI to be minimized is selected as:

$$J_M = \frac{1}{2} \int_0^{t_M} s^T(t + \tau) \cdot s(t + \tau) d\tau \tag{14}$$

One of the main results as well as the PID-CTOPC approach for the system (1) is given in Theorem 1.

**Theorem 1.** *Consider the nonlinear affine system (1) satisfying (A1)~(A3). By introducing a new PID-structuralized state variable shown in Formula (8), the prescribed reference tracking problem of the system can be transformed into a stabilization one given by (9). Then, for a given control order* $\rho \in \mathbb{Z}^+$*, the new state variable* $s(t)$ *is predicted by Formula (13). The optimal predictive control law minimizing the RHPI (14) is given by:*

$$u(t) = -[\overline{g}(x)]^{-1} \left[ \overline{f}(x) + K_M H_0 \right] \tag{15}$$

$K_M \in \mathbb{R}^{1 \times (n-1)}$ *is the first row of matrix* $B_\rho^{-1} B_1$*. The matrices* $B_1$ *and* $B_\rho$ *are given by:*

$$\begin{cases} B = \int_0^{t_M} \Gamma_M^T \Gamma_M d\tau = \begin{bmatrix} B_r & B_1^T \\ B_1 & B_\rho \end{bmatrix} \in \mathbb{R}^{(n+\rho) \times (n+\rho)} \\ B_r \in \mathbb{R}^{(n-1) \times (n-1)}, \quad B_\rho \in \mathbb{R}^{(\rho+1) \times (\rho+1)} \end{cases} \tag{16}$$

**Proof of Theorem 1.** Denote $\overline{u} = \left[ u, \dot{u}, \ldots, u^{[\rho]} \right]^T$, then the partial derivatives of $H_0$ and $H$ with respect to $\overline{u}$ can be given by:

$$\frac{\partial H_0}{\partial \overline{u}} = O^{(n-1) \times (\rho+1)}, \quad \frac{\partial H}{\partial \overline{u}} = \begin{bmatrix} \overline{g}(x) & 0 & \ldots & 0 \\ * & \overline{g}(x) & \ldots & 0 \\ * & * & \ddots & \vdots \\ * & * & * & \overline{g}(x) \end{bmatrix} \in \mathbb{R}^{(\rho+1) \times (\rho+1)} \tag{17}$$

where, $\partial H / \partial \overline{u}$ is a lower triangular matrix with the diagonal elements all $\overline{g}(x)$, and the symbol '*' represents the non-zero elements. It is obvious that the matrix $\partial H / \partial \overline{u}$ is full rank.

By considering Formula (13), differentiating $J_M$ with respect to $\overline{u}$ yields:

$$\frac{\partial J_M}{\partial \overline{u}} = \left[ \left( \frac{\partial H_0}{\partial \overline{u}} \right)^T, \left( \frac{\partial H}{\partial \overline{u}} \right)^T \right] \cdot B \cdot \begin{bmatrix} H_0 \\ H \end{bmatrix} \tag{18}$$

where, $B = \int_0^{t_M} \Gamma_M^T \Gamma_M d\tau$. Partition the matrix $B$ by following Formula (16). Then, according to the necessary condition $\frac{\partial J_M}{\partial \overline{u}} = 0$, the Formula (18) can be expanded as:

$$\left( \frac{\partial H}{\partial \overline{u}} \right)^T (B_1 H_0 + B_\rho H) = 0 \tag{19}$$

Recall that $\partial H / \partial \overline{u}$ is full rank, then formulation of $H$ can be given by:

$$H = -B_\rho^{-1} B_1 H_0 \tag{20}$$

Denoting $K_M \in \mathbb{R}^{1 \times (n-1)}$ as the first row of matrix $B_\rho^{-1} B_1$ and taking the first row of $H$ yields:

$$\overline{f}(x) + \overline{g}(x) \cdot u = -K_M H_0 \tag{21}$$

Then, the PID-CTOPC law can be given by:

$$u(t) = -[\overline{g}(x)]^{-1} \left[ \overline{f}(x) + K_M H_0 \right] \tag{22}$$

Proof of the theorem is accomplished. □

**Remark 1.** *Since the modified approach is an advanced version of the PCTOPC, hence, tuning principles of the parameter $t_M$ and the control order $\rho$ also follow the ones of the PCTOPC, which can be referred to [8]. Meanwhile, tuning of $K_p$, $K_i$, and $K_d$ also follow the principles of the conventional PID controller, namely, (1) large $K_p$ would accelerate the output response speed, which is beneficial to reduce the steady-state tracking error while increases the overshoot; (2) small $K_i$ would reduce the overshoot while slow down the speed of eliminating the steady-state error; and (3) large $K_d$ would accelerate the system response speed and reduce the overshoot while weakening the system robustness against disturbances.*

*3.3. Stability Analysis*

In this subsection, stability of the closed-loop system under the derived optimal predictive controller (15) is analyzed, which is given in Theorem 2.

**Theorem 2.** *Consider a class of SISO nonlinear affine system (1) satisfying (A1)~(A3) with system dimension $n \geq 2$. For the derived optimal predictive controller (15), there must exist a predictive period $t_M > 0$, PID gains $K_p$, $K_i$, and $K_d$ such that the closed-loop system is stable.*

**Proof of Theorem 2.** In the derived controller (15), the gain matrix $K_M$ can be expanded as $K_M = [K_0, K_1, \ldots, K_{n-1}]$. Then the expanded formulation of the term $K_M H_0$ can be given by:

$$\begin{aligned} K_M H_0 = &\; K_0 K_i \int_0^t e_1 dt_0 + (K_0 K_p + K_1 K_i) e_1 \\ &+ \sum_{j=1}^{n-2} (K_j K_p + K_{j+1} K_i + K_{j-1} K_d) e_1^{[j]} + (K_{n-1} K_p + K_{n-2} K_d) e_1^{[n-1]} \end{aligned} \tag{23}$$

Recall in system (1) that $y^{[n]} = \dot{x}_n = f(x) + g(x) \cdot u$, then bringing controller (15) into the formulation $y^{[n]}$ by considering Formula (23) yields:

$$\begin{aligned} 0 = &\; e_1^{[n]} + \frac{(K_{n-1} + 1)K_p + K_{n-2}K_d}{K_d} e_1^{[n-1]} + \frac{K_{n-2}K_p + (K_{n-1} + 1)K_i + K_{n-3}K_d}{K_d} e_1^{[n-2]} \\ &+ \frac{1}{K_d} \sum_{j=1}^{n-3} (K_j K_p + K_{j+1} K_i + K_{j-1} K_d) e_1^{[j]} + \frac{K_0 K_p + K_1 K_i}{K_d} e_1 + \frac{K_0 K_i}{K_d} \int_0^t e_1 dt_0 \end{aligned} \tag{24}$$

Differentiating both sides of Formula (24) with respect to time yields:

$$\begin{aligned} 0 = &\; e_1^{[n+1]} + \underbrace{\frac{(K_{n-1} + 1)K_p + K_{n-2}K_d}{K_d}}_{\beta_n} e_1^{[n]} + \underbrace{\frac{K_{n-2}K_p + (K_{n-1} + 1)K_i + K_{n-3}K_d}{K_d}}_{\beta_{n-1}} e_1^{[n-1]} \\ &+ \sum_{j=2}^{n-2} \left( \underbrace{\frac{K_j K_p + K_{j+1} K_i + K_{j-1} K_d}{K_d}}_{\beta_j} \right) e_1^{[j]} + \underbrace{\frac{K_0 K_p + K_1 K_i}{K_d}}_{\beta_1} \dot{e}_1 + \underbrace{\frac{K_0 K_i}{K_d}}_{\beta_0} e_1 \end{aligned} \tag{25}$$

Poles of the system (25) can be calculated according to the following characteristic equation:

$$e_1^{[n+1]} + \beta_n e_1^{[n]} + \beta_{n-1} e_1^{[n-1]} + \sum_{j=2}^{n-2} \beta_j e_1^{[j]} + \beta_1 \dot{e}_1 + \beta_0 e_1 = 0 \qquad (26)$$

Notice that the parameters $K_p$, $K_i$, $K_d$, and $K_j$, $j = 0, \ldots, n-1$ are all constants such that roots of the polynomial function (26) can be computed for different PID gains and predictive periods. Thus, appropriate values of $K_p$, $K_i$, $K_d$, and $K_j$ can make the system (25) stable. Then, proof of this theorem is fulfilled. $\square$

*3.4. Connections and Comparisons between the PCTOPC and the PID-CTOPC*

In this subsection, Figure 1 is applied to highlight both connections and differences between the PCTOPC and the proposed PID-CTOPC. Besides, flow chart for implementation of the PID-CTOPC is also illustrated in the figure.

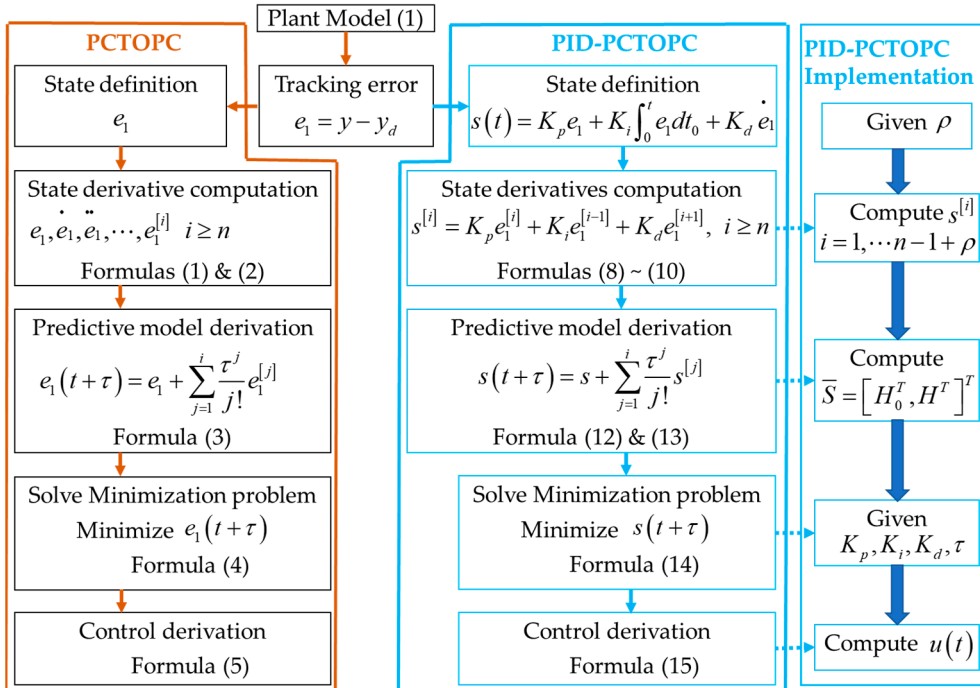

**Figure 1.** Connections and comparisons between the PCTOPC and the PID-CTOPC.

## 4. Model Simulation Results

This section aims to demonstrate the effectiveness of the PID-CTOPC approach numerically through designing autopilots for a missile under external disturbances. Comparison with the PCTOPC and the Lu's approach is carried out to highlight the superiorities of the PID-CTOPC. Lu's approach can enable the closed-loop system better robustness than the one based upon the PCTOPC and has made many achievements [34–45]. Introduction of Lu's approach targeting system (1) is given in Appendix A.

Longitudinal dynamics of a missile flying at Mach 3 and an altitude of 6096 m (20,000 ft) perturbed by disturbance is given by [19,40]:

$$\begin{cases} \dot{\alpha} = \frac{0.7 P_0 S}{m v_s} M_a C_n \cos \alpha + q \\ \dot{q} = \frac{0.7 P_0 S d_r}{I_y} M_a^2 C_m \\ \dot{\delta}_1 = \delta_2 \\ \dot{\delta}_2 = -\omega_a^2 \delta_1 - 2\xi_a \omega_a \delta_2 + \omega_a^2 \delta_c + d(t) \end{cases} \qquad (27)$$

where, $d(t)$ represents the unknown external disturbance. The force coefficient $C_n$ and the torque coefficient $C_m$ are given by:

$$\begin{cases} C_n = a_n\alpha^3 k^3 + b_n\alpha|\alpha|k^2 + c_n\left(2 - \frac{M_a}{3}\right)\alpha k + d_n k\delta_1 \\ C_m = a_m\alpha^3 k^3 + b_m\alpha|\alpha|k^2 + c_m\left(-7 + \frac{8M_a}{3}\right)\alpha k + d_m k\delta_1 \end{cases} \tag{28}$$

where, $\alpha$, $q$, $\delta_1$, and $\delta_2$ are system states. $\alpha$ and $q$ represent the attack angle (rad) and the pitch rate (rad/s), respectively. Differential equations of $\delta_1$ and $\delta_2$ describe the actuator dynamics. $\delta_c$ represents the tail deflection (rad) as well as the control input, which ranges from $-30$ degree to $30$ degree. The attack angle $\alpha$ is taken as the system output, which ranges from $-20$ degree to $20$ degree. Values of the model parameters in Formulas (27) and (28) can be seen in [19,46].

Two simplifications in [19] are adopted. Firstly, $\cos\alpha \approx 1$. Secondly, since the control surface mainly generates rotational torque, thus, its contribution in the force equation $C_n$, namely, the term $d_n k\delta_1$, is ignored. Hence, in the controller design, $\overline{C}_n = a_n\alpha^3 k^3 + b_n\alpha|\alpha|k^2 + c_n\left(2 - \frac{M_a}{3}\right)\alpha k$ is used to replace $C_n$. In addition, to write the missile dynamics into the form in system, let $x = [x_1, x_2, x_3, x_4]^T = [\alpha, q, \delta_1, \delta_2]^T$ and denote:

$$\begin{cases} [f_1(x), f_2(x), f_3(x), f_4(x)]^T = \left[\frac{0.7P_0S}{mv_s}M_a\overline{C}_n + q, \frac{0.7P_0Sd_r}{I_y}M_a^2 C_m, \delta_2, -\omega_a^2\delta_1 - 2\xi_a\omega_a\delta_2\right]^T \\ [g_1(x), g_2(x), g_3(x), g_4(x)]^T = [0, 0, 0, \omega_a^2]^T \end{cases} \tag{29}$$

Next, to design the controller, the unknown external disturbance $d(t)$ is assumed to be zero. Then, in the simulation, $d(t)$ is considered. Differentiating the attack angle $\alpha$ four times yields:

$$\begin{cases} \dot{\alpha} = f_1(x) \\ \ddot{\alpha} = \frac{\partial f_1(x)}{\partial x_1}f_1(x) + f_2(x) \\ \alpha^{[3]} = \left[\frac{\partial^2 f_1(x)}{\partial x_1^2}f_1(x) + \left(\frac{\partial f_1(x)}{\partial x_1}\right)^2 + \frac{\partial f_2(x)}{\partial x_1}\right]f_1(x) + \frac{\partial f_1(x)}{\partial x_1}f_2(x) + \frac{\partial f_2(x)}{\partial x_3}f_3(x) \\ \alpha^{[4]} = \rho_1(x)\cdot f_1(x) + \rho_2(x)\cdot f_2(x) + \frac{\partial f_1(x)}{\partial x_1}\cdot\frac{\partial f_2(x)}{\partial x_3}f_3(x) + \frac{\partial f_2(x)}{\partial x_3}f_4(x) + \frac{\partial f_2(x)}{\partial x_3}g_4(x)\cdot u \end{cases} \tag{30}$$

where, the nonlinear terms $\rho_1(x)$ and $\rho_2(x)$ are given by:

$$\begin{cases} \rho_1(x) = \frac{\partial f_1^3(x)}{\partial x_1^3}f_1^2(x) + \frac{\partial f_1^2(x)}{\partial x_1^2}\left[4f_1(x)\frac{\partial f_1(x)}{\partial x_1} + f_1(x) + f_2(x)\right] + 2\frac{\partial f_1(x)}{\partial x_1}\cdot\frac{\partial f_2(x)}{\partial x_1} + \left[\frac{\partial f_1(x)}{\partial x_1}\right]^3 \\ \rho_2(x) = 2f_1(x)\frac{\partial f_1^2(x)}{\partial x_1^2} + \left[\frac{\partial f_1(x)}{\partial x_1}\right]^2 + \frac{\partial f_2(x)}{\partial x_1} \end{cases} \tag{31}$$

It can be verified that $m(x) \neq 0$. Therefore, the system dimension is $n = 4$.

### 4.1. Case Study 1: Superioities Validation

In this subsection, two groups of simulations are carried out in MATLAB/Simulink with the simulation time 10 s and the sampling interval 0.002 s. The prescribed reference (unit: rad) is given by: $\alpha_d(t) = 0.3$ when $t \leq 2$, $\alpha_d(t) = -0.2$ when $2 \leq t \leq 5$, $\alpha_d(t) = 0.2$ when $5 \leq t \leq 8$, and $\alpha_d(t) = -0.1$ when $8 \leq t \leq 10$. Then, derivatives of $\alpha_d(t)$ are given by: $\dot{\alpha}_d(t) = \ddot{\alpha}_d(t) = \alpha_d^{[3]}(t) = \alpha_d^{[4]}(t) = 0$. Reference values of the controller parameters are given in Table 1.

**Table 1.** Values of controller parameters.

| Approach Name | Parameter Value |
|---|---|
| PID-CTOPC | $t_M = 0.5$, $\rho = 2$, $K_p = 10$, $K_i = 0.1$, $K_d = 0.05$ |
| PCTOPC | $t_C = 0.5$, $\sigma = 2$ |
| Lu's | $t_L = 0.08$, $Q_1 = 100$, $Q_2 = 0.2$, $Q_3 = Q_4 = 0$ |

Controller parameters of the three approaches are given based on the performance index that the autopilot system has almost the same response time. In the first group of simulation, the system is affected by constant disturbance $d(t) = 5$. Then, in the second group, time-varying disturbance $d(t) = 5 \sin(\pi t)$ is added to the system. The two types of disturbances are only used to show the superiorities of the modified approach, which may have differences with the real scenarios.

### 4.2. Case Study 2: Control Parameter Selection Validation

In this subsection, since tuning principles of $K_p$, $K_i$, and $K_d$ are the same with the conventional PID controllers, which has been pointed out before, thus, influences on the PID-CTOPC-based closed-loop performance from the control parameters $t_M$ and $\rho$ are mainly discussed. Values of $\rho$, $K_p$, $K_i$, and $K_d$ are the same with the ones shown in Table 1. In addition, to clearly show the influences, disturbances are not considered in next.

First, closed-loop performance based upon difference values of the predictive period, namely, $t_M = 1$, $t_M = 0.5$, and $t_M = 0.3$ is illustrated.

Second, closed-loop performance based upon difference values of the control order, namely, $\rho = 2$, $\rho = 3$, and $\rho = 4$ is illustrated.

### 5. Discussion

Contrast results for Case study 1 are given by Figure 2:

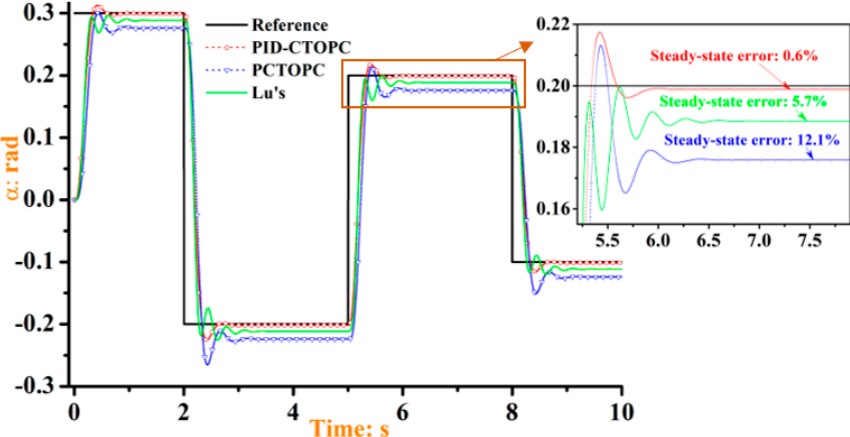

**Figure 2.** Group 1 with constant disturbance: Attack angle $\alpha$.

It can be seen from Figures 2–5 that, the PID-CTOPC approach is obviously superior to the two representative predictive approaches under the constant and time-varying external disturbances. For one thing, due to the consideration of the accumulative tracking error in the new error state, the proposed PID-CTOPC approach-based closed-loop system has smaller state-steady error than the one based on the PCTOPC and the Lu's approach, as shown in Figures 2 and 4. For another thing, using appropriate linear combination of the current, accumulative, and derivative tracking errors and minimizing the three errors simultaneously in the optimization problem are beneficial to reduce input fluctuations for the proposed approach compared with the PCTOPC and Lu's approach, which is also illustrated in Figures 3 and 5.

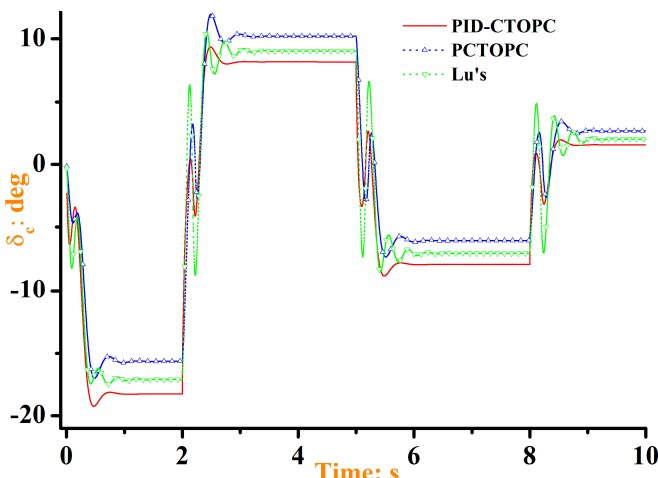

**Figure 3.** Group 1 with constant disturbance: Tail deflection $\delta_c$.

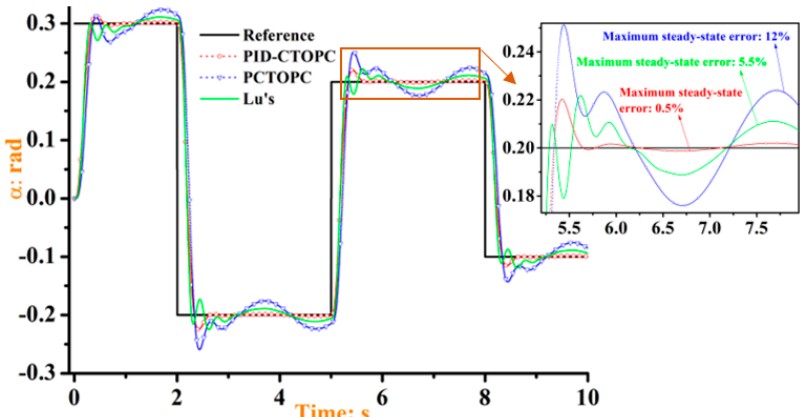

**Figure 4.** Group 2 with time-varying disturbance: Attack angle $\alpha$.

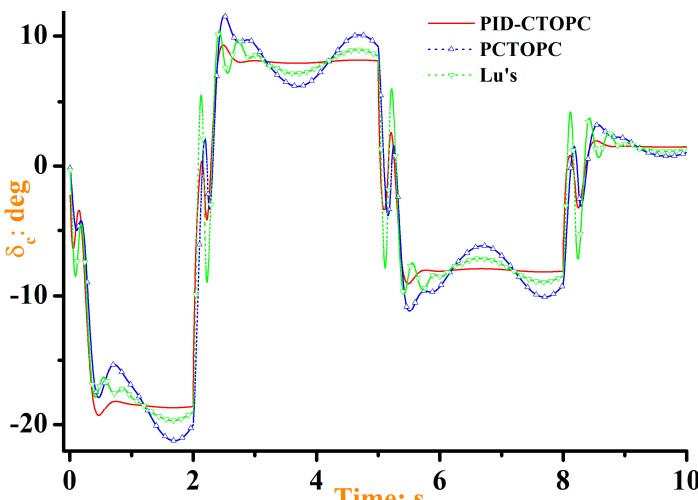

**Figure 5.** Group 2 with time-varying disturbance: Tail deflection $\delta_c$.

It can be seen from Figures 6 and 7 that a large value of the predictive period corresponds to fast system output response speed, while it would bring the closed-loop system obvious negative effects in which fluctuation and surging of the system input become serious in dealing with sudden changes. Thus, to derive satisfied control performance, one

should balance the response speed and the input fluctuation by adjusting the value of the predictive period.

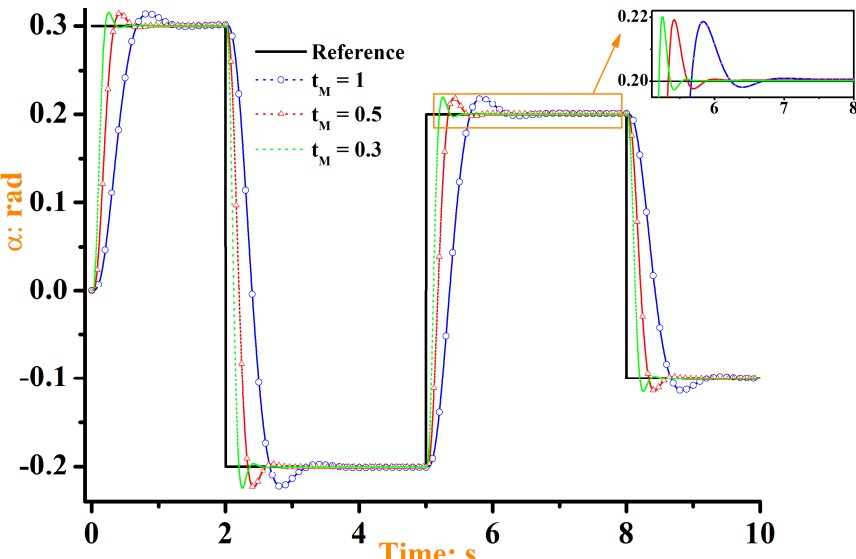

**Figure 6.** Validation with different predictive period $t_M$: Attack angle $\alpha$.

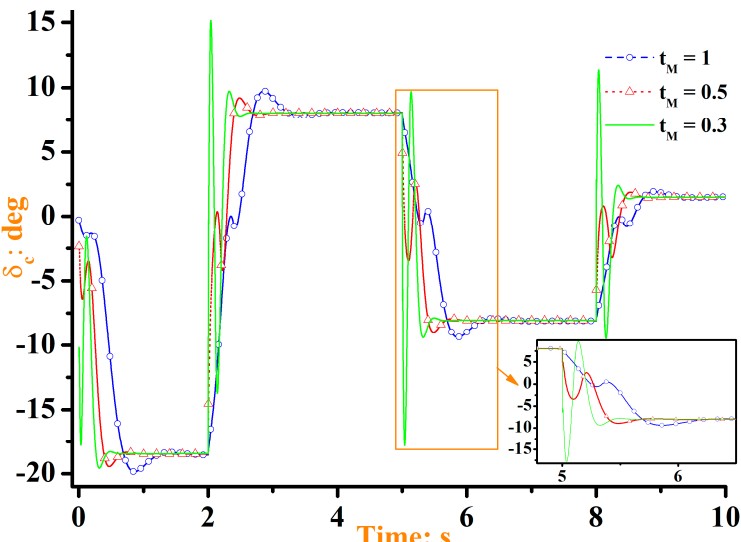

**Figure 7.** Validation with different predictive period $t_M$: Tail deflection $\delta_c$.

Figures 8 and 9 clearly illustrate that, though increasing the value of the control order would lead to small response time of the closed-loop system, frequent fluctuation and surging of the system input would occur when the sudden changes are encountered. Thus, tuning the value of the control order is another way to weigh the response speed and the input fluctuation for the closed-loop system.

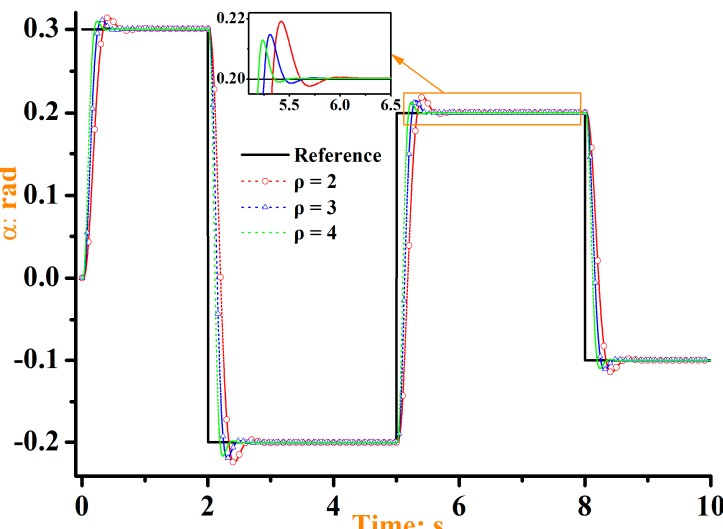

**Figure 8.** Validation with different control order $\rho$: Attack angle $\alpha$.

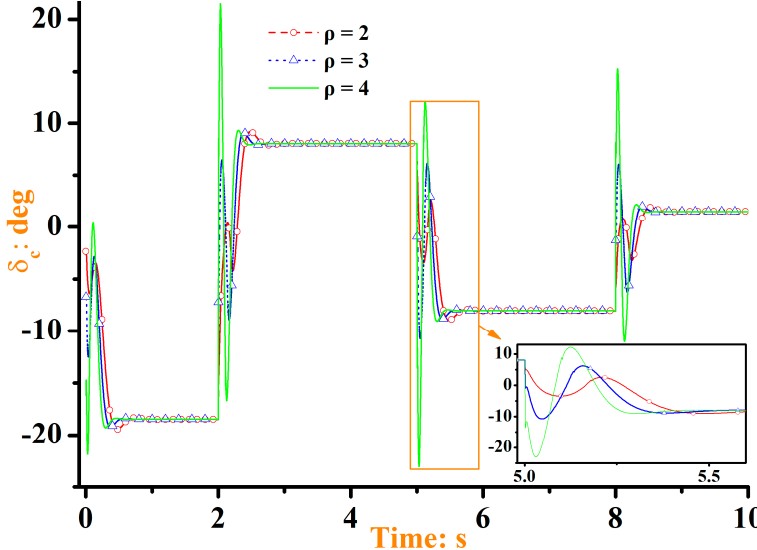

**Figure 9.** Validation with different control order $\rho$: Tail deflection $\delta_c$.

## 6. Conclusions

This paper improves the output performance for a class of high-order nonlinear systems under unknown constant and time-varying external disturbances by modifying the PCTOPC approach. The main conclusions are drawn as:

1.  A new system error state with PID structure is presented such that the current, the accumulative, and the derivative tracking errors are all considered and then minimized in the optimization problem simultaneously.

2.  Due to the minimization of the accumulative and derivative tracking errors in the proposed optimal predictive controller, robustness of the closed-loop system against the constant and the time-varying disturbances is significantly enhanced and the system input fluctuations are also degraded compared with the system using the controllers based upon the PCTOPC and the Lu's approaches.

3.  Though the proposed PID-CTOPC have achieved satisfactory control performance, there are some further studies that can be expected in the future. First, the proposed approach is expected to be validated experimentally. Second, the control parameters are required to be adjusted adaptively by using tracking errors.

**Author Contributions:** Conceptualization, G.W. and H.A.; methodology, G.W.; writing—original draft preparation, G.W.; writing, review and editing, G.W.; investigation, Q.L. All authors have read and agreed to the published version of the manuscript.

**Funding:** This research was funded by the Aeronautical Science Foundation of China, grant number F2021110/ ASFC-201913053001; the National Science Foundation of Shaanxi Province, grant number 2021JQ-078; Fundamental and Applied Fundamental Research of Zhuhai City, grant number ZH2201-7003-210011-P-WC and AG-EX-JSXY-009.

**Institutional Review Board Statement:** Not applicable.

**Informed Consent Statement:** Not applicable.

**Data Availability Statement:** The data presented in this study are available in https://github.com/wgl1981/weiyi.git.

**Conflicts of Interest:** The authors declare no conflict of interest.

## Appendix A

In this appendix, one of the commonly applied CTOPC approaches proposed by Professor Lu is introduced.

In system (1), by using the Taylor series expansion, predictions of the states $x_i(t)$, $i = 1, \ldots, n$ within the predictive period $t_L$ can be given by:

$$
\begin{cases}
x_1(t + t_L) \approx x_1(t) + t_L \cdot x_2(t) + \ldots + \frac{t_L^n}{n!}[f(x(t)) + g(x(t)) \cdot u(t)] \\
x_2(t + t_L) \approx x_2(t) + t_L \cdot x_3(t) + \ldots + \frac{t_L^{n-1}}{(n-1)!}[f(x(t)) + g(x(t)) \cdot u(t)] \\
\quad \vdots \\
x_n(t + t_L) \approx x_n(t) + t_L \cdot [f(x(t)) + g(x(t)) \cdot u(t)]
\end{cases}
\tag{A1}
$$

Similarly, predictions of the prescribed references for all states within $t_L$ can also be given by:

$$
\begin{cases}
x_1^d(t + t_L) \approx x_1^d(t) + t_L \cdot x_2^d(t) + \ldots + \frac{t_L^n}{n!}\dot{x}_n^d(t) = y_d(t) + t_L \cdot \dot{y}_d(t) + \ldots + \frac{t_L^n}{n!}y_d^{[n]}(t) \\
x_2^d(t + t_L) \approx x_2^d(t) + t_L \cdot x_3^d(t) + \ldots + \frac{t_L^{n-1}}{(n-1)!}\dot{x}_n^d(t) = \dot{y}_d(t) + t_L \cdot \ddot{y}_d(t) + \ldots + \frac{t_L^{n-1}}{(n-1)!}y_d^{[n]}(t) \\
\quad \vdots \\
x_n^d(t + t_L) \approx x_n^d(t) + t_L \cdot \dot{x}_n^d(t) = y_d^{[n-1]}(t) + t_L \cdot y_d^{[n]}(t)
\end{cases}
\tag{A2}
$$

Let $e_i(t) = x_i(t) - x_i^d(t)$, $i = 1, \ldots, n$ be the state tracking errors. Then, predictions of $e_i(t)$ can be expressed by:

$$
e_i(t + t_L) = x_i(t + t_L) - x_i^d(t + t_L)
\tag{A3}
$$

Denote $E(t + t_L) = [e_1(t + t_L), \ldots, e_n(t + t_L)]^T$. The RHPI to be minimized relative to the input $u(t)$ is selected as:

$$
J_L(u(t)) = \frac{1}{2}\left[E^T(t + t_L) \cdot M \cdot E(t + t_L) + \mu \cdot u^2(t)\right]
\tag{A4}
$$

where, $M \in \mathbb{R}^{n \times n}$ is a semi-definite diagonal weighting matrix with the diagonal elements $m_i \geq 0$, $i = 1, \ldots, n$. $\mu \in \mathbb{R}$ is a positive number called penalty factor.

Then, by using the necessary condition $\partial J_L(u(t))/\partial u(t) = 0$, the optimal predictive control law is given by:

$$
\begin{aligned}
u(t) = -\frac{1}{Q} \Bigg\{ & \frac{t_L^n}{n!} g(x) m_1 \left[ e_1(t) + t_L e_2(t) + \frac{t_L^2}{2!} e_3(t) + \ldots + \frac{t_L^n}{n!} \left( f(x) - y_d^{[n]}(t) \right) \right] \\
& + \frac{t_L^{n-1}}{(n-1)!} g(x) m_2 \left[ e_2(t) + t_L e_3(t) + \ldots + \frac{t_L^{n-1}}{(n-1)!} \left( f(x) - y_d^{[n]}(t) \right) \right] + \ldots \\
& + t_L g(x) m_n \left[ e_n(t) + t_L \left( f(x) - y_d^{[n]}(t) \right) \right] \Bigg\}
\end{aligned}
\tag{A5}
$$

where, $Q$ is given by:

$$
Q = g^2(x) \left[ \left( \frac{t_L^n}{n!} \right)^2 m_1 + \left( \frac{t_L^{n-1}}{(n-1)!} \right)^2 m_2 + \ldots + t_L^2 m_n \right] + \mu
\tag{A6}
$$

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
