# Peer review of "Linear Proportional-Integral-Differential-Robustified Continuous-Time Optimal Predictive Control for a Class of Nonlinear Systems"

_applsci, doi:10.3390/app12115446_

Round 1

Reviewer 1 Report

The paper proposes a robust optimal predictive control approach for a class of non-linear continuous-time systems perturbed by unknown disturbances. My comments/suggestions are given as follows.

  1. The main contributions of this paper should be further summarized and clearly demonstrated. why the proposed approach is needed to be used instead of the existing methods?
  2. Article simulations do not have clear Figs. Please add clear Figures to the article
  3. The connection between your work and the previous theory is not clear enough. More details are necessary for improved transparency.
  4. Time-domain or Frequency domain stability analysis should be added to the manuscript.
  5. Hardware implementation results should be included and compared.
  6. Sensitivity analysis should be added to the manuscript.
  7. How did you validate your giving simulation results?
  8. You should save the simulation program codes, and cite the link to them in the paper. This is useful for validation.

Reviewer 2 Report

Generally, the paper is well written and well organised,

I only have a few comments and questions:

Please highlight the motivation of this study and why such approach should be taken in practical applications compared to a vase majority of similar methods.

Most of the references in the literature review are old, it is necessary to show if the study is timely and new.

Provide a block diagram or a flow chart showing the steps of the controller design

Clarify the design variables and the parameters that should be selected prior to solving the set of equations.

The simulation section is very brief, expand on the impact of factors, for example discuss what how sensitive the results are to the design parameters?

Reviewer 3 Report

 This paper presents a novel robust optimal predictive control approach for a class of nonlinear continuous-time systems perturbed by unknown disturbances. In order to highlight superiorities of the proposed approach, comparison with another typical CTOPC approach which considers both current and high-derivative tracking errors is done by authors.

The Introduction and Preliminaries chapters prove good knowledge of the state of the art.

Chapter 3 presents main results of the paper. The equations are presented in successive order, based on the preliminary assumptions, theorems and proof of theorems being demonstrated.

The Illustrative Examples chapter is about validation and effectiveness of the PID-CTOPC approach. The plotted graphs are aimed to proof it.  

Suggestion: Change the name of this chapter. For example: Model simulation results.

Conclusion chapter, could be extended – for example mentioning further research development.

Check English and phrase – for example line 119 (“where”), line 181 (the sign)

Round 2

Reviewer 2 Report

I am happy with the responses